A novel microbial agent reduces soil paclobutrazol residue, enhances enzyme activities and increases Ophiopogon japonicus production

Jiang Xiaojun 1 2 3
Wang Huadong 1 2 3
Huang Yi 4
Jin Hong jh1120@139.com 1 2 3
Ding Jianguo 5
Ma Lin 4
Zheng Lei 1 2 3
1 School of Pharmacy, Sichuan College of Traditional Chinese Medicine , Mianyang , Sichuan , China
2 Northwest Sichuan Laboratory of Traditional Chinese Medicine Resources Research and Development Utilization , Mianyang , Sichuan , China
3 Mianyang Key Laboratory of Development and Utilization of Chinese Medicine Resources , Mianyang , Sichuan , China
4 College of Life Science and Engineering, Southwest University of Science and Technology , Mianyang , Sichuan , China
5 Mianyang Xintengyuan Traditional Chinese Medicine Co., Ltd. , Mianyang , Sichuan , China
Kutlu Imren
Electronic publication date: 2025 Feb 20
Publication date: 2025
Volume: 13
Electronic Location ID: e19008
Received 2024 Dec 10; Accepted 2025 Jan 27
Copyright: ©2025 Jiang et al.
Copyright year: 2025
Copyright holder: Jiang et al.
License: This is an open access article distributed under the terms of the Creative Commons Attribution License, which permits unrestricted use, distribution, reproduction and adaptation in any medium and for any purpose provided that it is properly attributed. For attribution, the original author(s), title, publication source (PeerJ) and either DOI or URL of the article must be cited.
License URL: https://creativecommons.org/licenses/by/4.0/

Keywords: Microbial agent, Paclobutrazol residue, Enzyme activity, Yield, Dry matter ratio, Ophiopogon japonicus

Funding: Teacher Key Research Project of Sichuan College of Traditional Chinese Medicine 23ZRZD01 Key Discipline Construction Project of Sichuan Provincial Administration of Traditional Chinese Medicine 2020ZD07 This work was supported by the Teacher Key Research Project of Sichuan College of Traditional Chinese Medicine (No. 23ZRZD01) and the Key Discipline Construction Project of Sichuan Provincial Administration of Traditional Chinese Medicine (No. 2020ZD07). The funders had no role in study design, data collection and analysis, decision to publish, or preparation of the manuscript.

==============================
Background

Ophiopogon japonicus (O. japonicus) is a versatile plant valued for its medicinal, food, and ornamental properties. Its cultivation often involves the excessive use of paclobutrazol, leading to a series of environmental and agricultural problems such as soil contamination, nutrient depletion, and safety risks. However, there is currently no effective solution.

Methods

Based on a novel microbial agent, Micrococcus yunnanensis strain HY001 (MYSH), field experiments were conducted in the main production area of O. japonicus. Soil paclobutrazol residue, soil enzyme activities, and the yield and dry matter ratio of O. japonicus were measured. Hierarchical partitioning (HP) was used to identify the relative importance of different variables, and partial least squares path modeling (PLS-PM) was applied to elucidate the mechanisms underlying MYSH’s effects on soil health and crop production.

Results

MYSH significantly reduced soil paclobutrazol residue by 75.18% over five months, compared to a natural degradation rate of 50.72% over a year. Compared to the control group, the MYSH-treated group enhanced soil sucrase activity, soil urease activity, and soil alkaline phosphatase activity, with rates of 47.81%, 46.70%, and 216.66%, respectively. Additionally, MYSH improved O. japonicus productivity, with a 94.75% increase in yield and a 17.64% increase in dry matter ratio. HP revealed that MYSH was the primary factor affecting the yield and dry matter ratio of O. japonicus, with relative importance of 47.75% and 42.28%, respectively. The key mechanism was that MYSH degraded soil paclobutrazol residue, which in turn influenced soil sucrase activity, ultimately impacting the yield of O. japonicus (p < 0.05).

Conclusions

This study demonstrates the dual role of MYSH as both an environmental remediation agent and a crop productivity enhancer for the first time. By reducing paclobutrazol residue and enhancing soil health and crop production, MYSH shows great potential for broader application in sustainable agricultural practices. This research highlights the efficacy of microbial agents in addressing agrochemical contamination and promoting sustainable farming, providing a valuable contribution to the development of eco-friendly agricultural solutions.

Introduction

Ophiopogon japonicus (L. f) Ker-Gawl., also known as Maidong in Chinese, is a versatile plant with multiple uses as a medicinal herb, food source, and ornamental plant (Liu et al., 2023a). It is commonly cultivated in East Asia, particularly in China, South Korea, and Japan (Chen et al., 2016). Phytochemical studies have identified various bioactive compounds in O. japonicus, including steroidal saponins, homoisoflavonoids, and polysaccharides (Liu et al., 2023a). These compounds have been shown to possess various biological activities, including cardiovascular protection, anti-inflammatory effects, anticancer properties, and antioxidant effects (Wang et al., 2017; Zhao et al., 2017; Lei, Weckerle & Heinrich, 2021). Due to its medicinal value, the demand for O. japonicus has increased significantly. To meet this demand, plant growth regulators like paclobutrazol are widely used in its cultivation (Zhang et al., 2021). However, whose excessive use can have negative environmental and agricultural consequences, such as soil nutrient decline and detrimental effects on other crops (Jiang et al., 2019). Furthermore, paclobutrazol residue may accumulate in medicinal materials, affecting its quality and safety for medicinal and food purposes (Wang, Wu & Lonameo, 2019). Therefore, there is a pressing need to develop agricultural practices that enhance soil fertility while minimizing the harmful effects of chemical residues, ensuring sustainable production of O. japonicus.

Microbial agents, derived from natural sources, have emerged as a promising alternative to chemical fertilizers and pesticides for improving soil health and enhancing crop productivity (Sahu et al., 2019; Kour et al., 2020; Arora et al., 2024). These microbial agents can modulate soil microbial communities, enhance soil enzyme activities, and promote nutrient cycling, thereby improving soil fertility and supporting plant growth (Sun et al., 2017; Deng et al., 2021; Ahsan et al., 2024). Furthermore, microbial agents have been shown to degrade various pesticides in the soil, including paclobutrazol, through biodegradation processes (Liu et al., 2023b; Shahid, Khan & Singh, 2023). Due to their safety and non-toxic nature, microbial agents have been applied in a range of agricultural practices, such as soil improvement, reduction of pesticide residues, and pest control (Marian & Shimizu, 2019; Datta et al., 2024). Thus, the application of microbial agents offers a potential solution to mitigate the negative impacts of paclobutrazol residue on soil health and the production of O. japonicus. Micrococcus yunnanensis sp. nov., an actinobacterium isolated from the roots of Polyspora axillaris in 2009, has shown significant potential in various biotechnological applications (Zhao et al., 2009). For instance, carotenoids and its derivatives have been isolated from Micrococcus yunnanensis strains, with antioxidative properties (Osawa et al., 2010). Another strain of Micrococcus yunnanensis, isolated from Sarcheshme mine soil, has been found to facilitate the biosynthesis of gold nanoparticles, which exhibited toxic effects on cancer cell lines and antibacterial activity against various pathogens in vitro (Jafari et al., 2018).

Soil enzyme activities are regarded as integrative bioindicators because they reveal the composition and roles of microbial populations. As a result, soil enzyme activities are frequently employed to track how agricultural practices, environmental contaminants, and soil management affect soil health (Yuan et al., 2017). Pollutants may change nutrient cycling, plant growth and crop yield if they affect soil enzyme activities (Molaei et al., 2017). Consequently, soil health can be evaluated through measurements of enzyme activity. The enzymes of interest are soil sucrase, urease activity, and soil alkaline phosphatase in this study. Soil sucrase catalyzes the hydrolysis of sucrose into glucose and fructose and can be used to assess soil fertility and nutrient utilization (Zhou et al., 2022). Soil urease facilitates the transformation of organic nitrogen to ammonia in soil, thus providing essential nutrients to plants (Kompała-Bąba et al., 2021). Soil alkaline phosphatase are one of the predominant forms across a wide range of soil pH conditions and play an important role in bringing nutrient supply more closely in line with nutrient demand (Rocabruna et al., 2024). Therefore, due to the presence of paclobutrazol, any positive or negative influence on these soil enzymes, could influence soil health and crop production.

This study built on our previous work, where we identified and patented the Micrococcus yunnanensis strain HY001 (MYSH) isolated from soils in O. japonicus cultivation areas treated with paclobutrazol (Ding et al., 2023). Preliminary results revealed MYSH’s ability to degrade paclobutrazol residue. Here, we presented the first comprehensive field investigation into the effects of MYSH on soil paclobutrazol residue, soil enzyme activities, and the production of O. japonicus. By employing rigorous field trials using a randomized block design and applying advanced statistical tools such as HP and PLS-PM, we elucidated the mechanisms underlying the interactions between MYSH, soil health, and crop productivity. This novel approach contributes to the development of sustainable agricultural practices that address the dual challenges of environmental sustainability and high-quality production of O. japonicus.

Materials & Methods

Chemicals and reagents

The standard stock solutions (100 mg/L) of paclobutrazol were purchased from the Tanmo Quality Control Technology Co. Ltd. (Beijing, China), and the wettable powder of 15% effective content paclobutrazol was purchased from Sichuan Runer Technology Co., Ltd. (Chengdu, China). 3,5-dinitrosalicylic acid and disodium phenyl phosphate were purchased from Chengdu Kelong Chemical Co., Ltd. Acetonitrile (chromatographically pure) was purchased from Thermo Fisher Scientific Inc. (Shanghai, China). In 2022, a variety of microorganisms were isolated and purified from soil samples collected in the O. japonicus cultivation region of Santai County, Mianyang City, Sichuan Province, China. Using spread plate and streak plate methods, the strains were identified through the polymerase chain reaction (PCR), basic local alignment search tool (BLAST), and phylogenetic analysis.

MYSH was inoculated into LB medium (containing 12 g/L tryptone, six g/L yeast extract, and eight g/L NaCl) and cultured at 30 °C, 150 rpm, until the OD600 value of the bacterial solution was 0.6–0.8, and the seed bacterial solution was obtained. The 50 mL seed solution of MYSH was added to a 50 L fermenter (containing 70 g/L soybean powder, 70 g/L starch, three g/L sodium chloride, 3.5 g/L calcium carbonate, 5.2 g/L potassium dihydrogen phosphate, and five g/L ferrous sulfate) and cultured at 30 °C until the number of viable bacteria reached 109 CFU/g (Wang et al., 2024). The culture obtained was the microbial agent used in this study. Meanwhile, the MYSH was deposited in the China General Microbiological Culture Collection Center (CGMCC) on November 11, 2022, with the preservation number of CGMCC No.: 26130.

Sampling point and experimental design

The test base is located in Santai County (Fig. 1), Mianyang City, Sichuan Province, China (31°17′24″N, 104°56′14″E), which is the main production area of O. japonicus, with a stable planting area of 4,000 ha, accounting for more than 95% of China’s exports (Lyu et al., 2020). It is a subtropical humid monsoon climate zone with an average annual temperature of 16–17 °C. The annual average sunshine hours are ≥ 1,260 h, rainfall is 850–900 mm, the frost-free period is ≥ 275 d (https://data.cma.cn/en), and the soil type is sandy soil (http://soil.geodata.cn).

O. japonicus was cultivated with two tillers per hole at a density of 120 plants/m2 on 20 April 2023. The dose of paclobutrazol and MYSH solution was used based on the previous study (Zhang et al., 2021; Ding et al., 2023), and the experimental group and treatment were shown in Table 1. On 10 November 2023, the clean water (control group), paclobutrazol solution, MYSH solution, and paclobutrazol & MYSH solution were uniformly sprayed on the leaves of plants and soil with small sprayers at a height of about 1 m above the ground (Zhang et al., 2021). There was no rainfall within three days before and after spraying. Each experimental group contained three randomized experimental districts, each district was 3 m2, and a 0.5 m buffer zone was maintained between experimental groups. Before planting O. japonicus in April 2023, 15,000 kg/ha of farmyard manure and 600 kg/ha of nitrogen, phosphorus and potassium compound fertilizer (total nutrient ≥ 45%, nitrogen: potassium: potassium = 15:15:15) were used. In May and August 2023, the same compound fertilizer with 225 kg/ha was used. In October 2023, 600 kg/ha of potassium chloride fertilizer was used. Irrigation was carried out according to the season, climate and soil conditions, without the use of any pesticides during cultivation. When harvested on 20 April 2024, all the tuberous roots of five O. japonicus were randomly collected in each experimental district. At the same time, the soil around the tuberous roots was collected, the visible debris was removed, and dried naturally.

Figure 1 Landmasses of the study area.

(1) Control group, (2) paclobutrazol treatment group, (3) Micrococcus yunnanensis strain HY001 treatment group, (4) paclobutrazol and Micrococcus yunnanensis strain HY001 treatment group. The data comes from the Map Data ©2022 Google.

Table 1 The experimental group and treatment in this study.

No.	Experimental group	Experimental treatment	
1	Control group	12.5 L of clean water	
2	Paclobutrazol	10 L of solution containing 100 g of paclobutrazol wettable powder + 2.5 L of clean water	
3	MYSH	2.5 L of MYSH +10 L of clean water	
4	Paclobutrazol & MYSH	10 L of solution containing 100 g of paclobutrazol wettable powder + 2.5 L of MYSH	
Notes.

MYSH represents Micrococcus yunnanensis strain HY001.

Evaluation of soil paclobutrazol residue

The soil paclobutrazol residue was evaluated based on the method described in the Chinese Pharmacopoeia Commission (National Pharmacopoeia Commission, 2020). Namely, it was described as high performance liquid chromatography-tandem mass spectrometry (HPLC-MS/MS), with the limit of quantitation (LOQ) of 0.01 mg/kg. The air-dried soil samples (including before and after O. japonicus cultivation) were ground and passed through a sieve with an aperture of 355 µm ± 13 µm. About 3.0 g of soil powder was placed in a 50 mL polystyrene plug centrifuge tube, 15 mL of acetonitrile was added, and it was placed on an oscillator for violent oscillation (2,000 times/min) for 20 min, centrifuged at 5,000 r/min for 5 min, and filtered through a 0.22 µm filter membrane (National Pharmacopoeia Commission, 2020).

Determination of soil sucrase, urease and alkaline phosphatase activity

The air-dried soil samples was passed through a two mm sieve. For soil sucrase activity determination, five g of soil was placed in a 50 mL triangular flask. After the addition of five drops of toluene, the mixture was allowed to stand for 15 min. Subsequently, 15 mL of 8% (weight: volume) sucrose solution and five mL of pH 5.5 phosphate buffer were injected, and the mixture was thoroughly shaken. The flask was then incubated at 37 °C for 24 h. One milliliter of the filtrate was pipetted into a 50 mL volumetric flask, followed by the addition of three mL of 3,5-dinitrosalicylic acid. The flask was heated in boiling water for 5 min and immediately transferred to running tap water to cool for 3 min. Finally, the solution was diluted to 50 mL with distilled water and was measured spectrophotometrically at 508 nm. Soil sucrase activity was expressed as milli-grams of glucose generated by per gram of dry soil in 24 h (Yuan et al., 2017).

For soil alkaline phosphatase activity, five g of soil was placed in a 200 mL triangular flask. Subsequently, 2.5 mL of toluene was added, and the mixture was gently shaken for 15 min. Then, 20 mL of a 0.5% solution of disodium phenyl phosphate was introduced into the flask. The pH was adjusted and maintained in an alkaline range using a borate buffer. After thorough mixing, the flask was incubated at 37 °C for 24 h. Finally, the supernatant was collected, and its absorbance was measured at 660 nm. Its activity was expressed as milligrams of phenol generated by per gram of dry soil over 24 h (Li et al., 2021).

Regarding soil urease activity, it was determined using the method described by a previous study. Five g of soil was placed in a 50 mL triangular flask, then one mL of toluene was added, and after shaking evenly, 15 min after adding 10 mL (c = 0.1 g/mL) urea solution and 20 mL (pH = 6.7) citrate buffer solution, it was shaken well in a 37 °C incubator for 24 h. After the culture was filtered, one mL of filtrate was added to a 50 mL volumetric flask, and then four mL of phenol sodium solution and three mL of sodium hypochlorite solution were added and shaken well. After 20 min, the color was developed and the volume was fixed. The absorbance was measured with a spectrophotometer at a wavelength of 578 nm within 1 h. The urease activity was expressed as milli-grams of NH3–N released per gram of dry soil over 24 h (Fernandez-Calvino et al., 2010).

Measurement of the yield and dry matter ratio of O. japonicus

The tuberous roots were washed separately and dried at 105 °C for 15 min. The dry weight was recorded until they reached a constant weight at 60 °C. The yield of O. japonicus was converted according to planting density and calculated using the following formula (Lei et al., 2019): Yieldt/ha=Wdryn×80,000×151,000×1,000

where Wdry is the dry weight of O. japonicus tuberous roots and n is the number of plants in each experimental group. The dry matter ratio of O. japonicus was calculated using the following formula (Ucar et al., 2017): Dry matter ratio%=WdryWfresh×100

where Wdry is the dry weight of O. japonicus tuberous roots and Wfresh is the fresh weight of O. japonicus tuberous roots.

Statistical analysis

Using SPSS 24.0, the Shapiro–Wilk test was used to assess normality before testing for significance, the homogeneity of variance was analyzed with an F-test, and the one-way analysis of variance (ANOVA) was conducted to assess the differences in soil enzyme activities, soil paclobutrazol residue, and the yield and dry matter ratio of O. japonicus. The Spearman correlation was employed to analyze their associations because there were both scale variables and ordinal variables, which were categorized based on the measure of SPSS 24.0. The HP was performed using the “glmm.hp” package in R version 4.4.0 (Lai et al., 2022; Lai et al., 2023).

Partial least squares structural equation modeling is a regression-based modeling method that utilizes a principal component-centric approach to describe the direct dependencies between a range of variables, which has been used in several studies (Hair & Alamer, 2022). In this study, PLS-PM was used to estimate the relationships between MYSH, soil paclobutrazol residue, and soil enzyme activities using the “plspm” package in R version 4.4.0 (Henseler, 2017). A bootstrap method (1,000 iterations) was used to validate the estimates of the path coefficients and the coefficients of determination (R2) (Tang et al., 2023). All figures were drawn using the GraphPad Prism (GraphPad Software 8.0.1) and “ggplot2” package in R version 4.4.0 (Swift, 1997; Villanueva & Chen, 2019).

Results

Soil paclobutrazol residue

Soil paclobutrazol residue before and after O. japonicus cultivation was evaluated, and the results are shown in Fig. 2. Soil paclobutrazol already existed in the soil be-fore O. japonicus cultivation (30 March 2023), with a residue of 0.548 mg/kg, and the paclobutrazol residue in the control group, paclobutrazol group, MYSH group, and paclobutrazol & MYSH group after cultivation of O. japonicus were 0.270 mg/kg, 2.540 mg/kg, 0.067 mg/kg, and 1.270 mg/kg, respectively. The soil paclobutrazol residue in the paclobutrazol group was significantly higher than that of the other two treatment groups and control group (p < 0.05); that of the paclobutrazol & MYSH group was significantly higher than that of the other MYSH group and control group (p <  0.05), and that of the control group was significantly higher than that in the MYSH group (p < 0.05). Compared with the soil before O. japonicus cultivation, the soil paclobutrazol residue in the control group decreased by 0.278 mg/kg under natural conditions. MYSH group decreased by 0.203 mg/kg compared with control group, and paclobutrazol & MYSH group decreased by 1.270 mg/kg compared with paclobutrazol group.

Figure 2 Soil paclobutrazol residue before cultivation and four groups.

Data show the mean value ± standard deviation (n = 3). MYSH represents Micrococcus yunnanensis strain HY001. Different lowercase letters indicate significant differences in different groups (p < 0.05).

Soil sucrase, urease and alkaline phosphatase activity

In this study, we evaluated the impact of four groups on soil enzyme activities, including soil sucrase, urease, and alkaline phosphatase. The results indicated that the activity of soil sucrase was highest in the paclobutrazol & MYSH group, with an activity level of 53.19 mg/(g 24 h), significantly surpassing that of the other group (p < 0.05, Fig. 3). In contrast, the control group exhibited the lowest soil sucrase activity, with an activity level of 22.58 mg/(g 24 h). Regarding soil urease activity, the MYSH group demonstrated the highest activity, while the control group showed the lowest activity (Fig. 3), with an activity level of 19.72 mg/(g 24 h) and 6.23 mg/(g 24 h), respectively. There were significant differences in four groups (p < 0.05). For soil alkaline phosphatase activity, the MYSH group showed the highest activity with an activity level of 1.32 mg/(g 24 h), which was significantly higher than that of the other groups (p < 0.05), particularly the paclobutrazol group, which had the lowest activity with an activity level of 0.42 mg/(g 24 h) (Fig. 3). Compared to the control group, the MYSH-treated group enhanced soil sucrase activity, soil urease activity, and soil alkaline phosphatase activity, with rates of 47.81%, 46.70%, and 216.66%, respectively.

Figure 3 Soil sucrase, urease, and alkaline phosphatase activity in four groups.

Data show the mean value ± standard deviation (n = 3). MYSH represents Micrococcus yunnanensis strain HY001. Different lowercase letters indicate significant differences in different groups (p < 0.05).

The yield and dry matter ratio of O. japonicus

The yield and dry matter ratio of O. japonicus were measured, and the result was shown in Fig. 4. The yield of O. japonicus showed a gradual increase in control group, paclobutrazol, MYSH, and paclobutrazol & MYSH groups with values of 2.55 t/ha, 3.97 t/ha, 4.96 t/ha, and 7.14 t/ha, respectively. There were significant differences in the yield of O. japonicus in four groups (p < 0.05). Regarding the dry matter ratio of O. japonicus, they were 24.15%, 22.78%, 28.41% and 26.39% in the control group, paclobutrazol, MYSH, and paclobutrazol & MYSH groups, respectively. Similarly, there were significant differences in the dry matter ratio of O. japonicus in four groups (p < 0.05). Overall, compared to the control group, the MYSH-treated O. japonicus improved productivity, with a 94.75% increase in yield and a 17.64% increase in dry matter ratio.

Figure 4 The yield and dry matter ratio of O. japonicus in four groups.

Data show the mean value ± standard deviation (n = 15). MYSH represents Micrococcus yunnanensis strain HY001. Different lowercase letters indicate significant differences in different groups (p < 0.05).

The main factors influencing the yield and dry matter ratio of O. japonicus

The results of the correlation analysis of each variable were shown in Table 2. The correlation coefficients between yield and dry matter ratio, soil sucrase activity, and MYSH were 0.59 (p < 0.05), 0.72 (p < 0.01), and 0.87 (p < 0.01), respectively. The correlation coefficients between dry matter ratio and paclobutrazol residue, soil alkaline phosphatase activity, and MYSH were −0.76 (p < 0.01), 0.69 (p < 0.05), and 0.87 (p < 0.01), respectively. The correlation coefficients between paclobutrazol residue and soil alkaline phosphatase activity was −0.92 (p < 0.01).

Table 2 The Spearman correlation coefficients for each variable.

	Yield	DMR	PR	SA	APA	UA	MYSH	
Yield	1.00							
DMR	0.59*	1.00						
PR	0.01	−0.76**	1.00					
SA	0.72**	0.12	0.48	1.00				
APA	0.03	0.69*	−0.92**	−0.53	1.00			
UA	0.37	0.46	−0.26	0.15	0.19	1.00		
MYSH	0.87**	0.87**	−0.44	0.44	0.44	0.44	1.00	
Notes.

DMR represents dry matter ratio. PR represents soil paclobutrazol residue. SA represents soil sucrase activity. APA represents soil alkaline phosphatase activity. UA represents soil urease activity. MYSH represents Micrococcus yunnanensis strain HY001.

* represents p < 0.05

** represents p < 0.01

The relative importance of each variable on the yield of O. japonicus is shown in Fig. 5. The relative importance of MYSH on the yield of O. japonicus was the highest, reaching 47.75%, followed by soil sucrase activity, which was 38.88%. The relative importance of soil paclobutrazol residue, soil urease activity, and soil alkaline phosphatase activity on the yield of O. japonicus were less than 6%, which were 5.28%, 4.23%, and 3.86%, respectively. These variables explained an R2 of 0.97 for the yield of O. japonicus. The relative importance of each variable on the dry matter ratio of O. japonicus was also shown in Fig. 5. The relative importance of MYSH on the dry matter ratio of O. japonicus was the highest, reaching 42.28%, followed by soil paclobutrazol residue and soil alkaline phosphatase activity, which was 21.85% and 20.15%, respectively. The relative importance of soil sucrase activity and soil urease activity on the dry matter ratio of O. japonicus were 6.76% and 8.96%, respectively. These variables explained an R2 of 0.93 for the dry matter ratio of O. japonicus.

Figure 5 The relative importance of different variables on the yield and dry matter ratio in O. japonicus.

MYSH represents Micrococcus yunnanensis strain HY001.

The relationship among soil paclobutrazol residue, enzyme activities, and yield and dry matter ratio of O. japonicus

Based on the PLS-PM, the interaction of the main influencing factors of O. japonicus yield and dry matter ratio was shown in Fig. 6. The yield of O. japonicus was directly stimulated by MYSH, soil paclobutrazol residue, and soil sucrase activity, whose path coefficients were 0.65 (p < 0.05), 0.20, and 0.68 (p < 0.05), respectively (R2 = 88%). Soil sucrase activity was directly stimulated by MYSH and soil paclobutrazol residue, whose path coefficients were 0.90 (p < 0.01) and 1.25 (p < 0.01), respectively (R2 = 73%). Notably, MYSH had a direct negative effect (p < 0.01) on soil paclobutrazol residue with a path coefficient of −0.73, showing the degradation effect of soil paclobutrazol by MYSH (R2 = 53%, Fig. 6A). Regarding the dry matter ratio of O. japonicus (Fig. 6B), it was directly stimulated by MYSH, soil paclobutrazol residue, and soil alkaline phosphatase activity, whose path coefficients were 1.03 (p < 0.01), 0.11, and 0.03, respectively (R2 = 95%). Soil alkaline phosphatase activity was directly stimulated by MYSH, whereas it was directly inhibited by soil paclobutrazol residue, whose path coefficients were 0.29 and −0.63 (p < 0.05), respectively (R2 = 76%). Similarly, this model also showed the degradation effect of soil paclobutrazol by MYSH (R2 = 53%).

Figure 6 The interaction among yield, dry matter ratio, Micrococcus yunnanensis strain HY001, soil enzyme activities and soil paclobutrazol residue.

(A) Direct effects for the yield of O. japonicus. (B) Direct effects for the dry matter ratio of O. japonicus. The ovals represent different variables, the arrows represent the link between them, and the numbers next to the arrows indicate path coefficients. The red lines indicate positive effects, and the blue lines indicate negative effects. MYSH represents Micrococcus yunnanensis strain HY001. An asterisk (*) represents p < 0.05, two asterisks (**) represent p < 0.01.

The analysis results of indirect effects and total effects are shown in Fig. 7. In the yield model of O. japonicus (Fig. 7A), the indirect effect of MYSH on soil paclobutrazol residue was 0, and the total effect was −0.73. The indirect effect on soil sucrase activity was −0.92, and the total effect was −0.02. The indirect effect on the yield of O. japonicus was −0.16, and the total effect was 0.50. The indirect effect of soil paclobutrazol residue on soil sucrase activity was 0, and the total effect was 1.25. The indirect effect on the yield of O. japonicus was 0.85, and the total effect was 1.05. The indirect effect of soil sucrase activity on the yield of O. japonicus was 0, and the total effect was 0.68. In the dry matter ratio model of O. japonicus (Fig. 7B), the indirect effect of MYSH on soil paclobutrazol residue was 0, and the total effect was −0.73. The indirect effect on soil alkaline phosphatase was 0.46, and the total effect was 0.75. The indirect effect on the dry matter ratio of O. japonicus was −0.06, and the total effect was 0.97. The indirect effect of soil paclobutrazol residue on soil alkaline phosphatase activity was 0, and the total effect was −0.63. The indirect effect on the dry matter ratio of O. japonicus was −0.02, and the total effect was 0.09. The indirect effect of soil alkaline phosphatase activity on the dry matter ratio of O. japonicus was 0, and the total effect was 0.03. Notably, the total effect of MYSH was manifested as reducing soil paclobutrazol residue, increasing soil enzyme activities and the yield of O. japonicus.

Figure 7 The indirect effects and total effects of soil paclobutrazol residue, soil enzyme activities and Micrococcus yunnanensis strain HY001.

(A) Indirect effects and total effects for the yield of O. japonicus. (B) Indirect effects and total effects for the dry matter ratio of O. japonicus. MYSH represents Micrococcus yunnanensis strain HY001. PR represents soil paclobutrazol residue. SA represents soil sucrase activity. APA represents soil alkaline phosphatase activity. DMR represents dry matter ratio. Direct effects + indirect effects = total effects.

Discussion

The establishment conditions and quantitative process for the HP and PLS-PM

Quantifying the influences of MYSH, soil paclobutrazol residue, and soil enzyme activities on the yield and dry matter ratio of O. japonicus is essential for identifying key factors that drive sustainable production. Our previous work demonstrated that MYSH had a degradation effect on soil paclobutrazol residue (Ding et al., 2023). In this context, the application of HP and PLS-PM enabled a quantitative assessment of the effects of these factors on both the yield and dry matter ratio of O. japonicus. However, few studies have quantitatively evaluated these influences due to the multicollinearity arising from the strong correlations between the factors, which can introduce significant errors into the evaluation (Table 2). Fortunately, the recently developed “glmm.hp” package addresses this issue effectively by providing a robust method for quantifying relative importance. This tool has gained widespread use in various research areas as a powerful statistical technique for isolating the individual contributions of different variables (Hu et al., 2023; Bibi et al., 2024; Li et al., 2024; Ni et al., 2024; Teng et al., 2024). In our study, the “glmm.hp” package was employed to quantify the relative importance of MYSH, soil paclobutrazol residue, and soil enzyme activities on the yield and dry matter ratio of O. japonicus (Fig. 5).

While HP using the “glmm.hp” package allows for the identification of the relative importance of different variables, it does not account for the direct and indirect interactions between them. In contrast, structural equation modeling (SEM) offers the advantage of quantifying causal relationships among multiple variables, but it is often prone to inaccuracies when dealing with large numbers of variables (Fan et al., 2016). Currently, SEM is typically categorized into two types: one based on maximum likelihood estimation and another based on partial least squares estimation. The latter, particularly PLS-PM, has several advantages, such as lower sample size requirements and greater suitability for analyzing newly constructed models (He et al., 2022; Liu et al., 2024). In this study, based on the preliminary results of HP, we applied PLS-PM to further analyze the interactions among MYSH, soil paclobutrazol residue, soil enzyme activities, and their impacts on the yield and dry matter ratio of O. japonicus (Fig. 6). The results obtained from this model not only clarified the most influential factors but also provided deeper insights into their underlying mechanisms. This analysis is a crucial step toward enhancing the production processes of O. japonicus and optimizing green production strategies.

The key mechanism underlying the yield and dry matter ratio of O. japonicus

The mechanisms underlying the yield of O. japonicus can be summarized in two categories: direct effects and indirect effects. On the one hand, MYSH, soil paclobutrazol residue, and soil sucrase activity directly impacted the yield of O. japonicus (Fig. 6A). On the other hand, MYSH indirectly affected yield by influencing soil paclobutrazol residue as well as by affecting soil sucrase activity. Meanwhile, MYSH affected soil paclobutrazol residue, which influenced soil sucrase activity and ultimately yield (Fig. 6A). Among these interactions, the most significant pathways were MYSH ->  soil sucrase activity -> the yield of O. japonicus and MYSH -> soil paclobutrazol residue -> soil sucrase activity -> the yield of O. japonicus (p < 0.05). Similarly, the mechanisms underlying the dry matter ratio of O. japonicus can also be classified into direct and indirect effects. Direct effects were observed from MYSH, soil paclobutrazol residue, and soil alkaline phosphatase activity, which directly influenced the dry matter ratio (Fig. 6B). Indirect effects were observed when MYSH influenced the dry matter ratio by affecting soil paclobutrazol residue and soil alkaline phosphatase activity. Additionally, MYSH affected soil paclobutrazol residue, which in turn influenced soil alkaline phosphatase activity, ultimately affecting the dry matter ratio (Fig. 6B).

In general, paclobutrazol is primarily recognized as a plant growth regulator that modifies the hormonal balance in crops, leading to increased yield (Davis, Curry & Steffens, 1991). However, deeper research has revealed additional effects in plants, including smaller stomatal pores, thicker leaves, increased surface appendage density, and higher root density (Rademacher, 2015). Furthermore, the impact of paclobutrazol on different soil enzymes is variable. Our results indicated that paclobutrazol had a significant positive effect on soil sucrase activity but a significant negative effect on soil alkaline phosphatase activity (Fig. 6), which is consistent with previous reports (Ren et al., 2021). In addition, paclobutrazol also had a significant negative effect on MYSH. It is due to its capacity as a triazole to inhibit sterol biosynthesis, which makes paclobutrazol have fungicidal activity (Rademacher, 2015). As in the previous study, paclobutrazol could increase the yield of O. japonicus tuberous roots (Davis, Curry & Steffens, 1991), but the effect on improving the dry matter ratio of O. japonicus was limited (Fig. 6). We infer that this may be due to paclobutrazol increasing the water content of O. japonicus tuberous roots while increasing the dry matter of O. japonicus tuberous roots, and the degree of water increase exceeds the dry matter. Consequently, although paclobutrazol enhances tuberous root yield, it results in a reduced dry matter ratio of O. japonicus

Comparative efficacy of MYSH and other microbial agents for soil remediation and crop enhancement

Microbial agents are increasingly recognized for their potential to enhance soil health and crop productivity, offering a sustainable alternative to chemical inputs (Kour et al., 2020). Research has demonstrated that microbial consortia can positively influence soil microbial communities by reducing pathogenic fungi and promoting plant growth (Ahsan et al., 2024). Targeted delivery systems, such as bionanofertilizers, further improve nutrient uptake and stress tolerance, thereby diminishing reliance on chemical fertilizers (Arora et al., 2024). Specific microbial strains capable of degrading persistent agrochemicals, including atrazine-degrading bacteria identified by Liu et al. (2023b), have been explored for their remediation potential. Additionally, the combined application of microbial agents and plants has shown beneficial effects on soil properties, as evidenced by improved nutrient content and enzyme activity in Pisha sandstone soils (Deng et al., 2021).

Among these microbial agents, MYSH has emerged as a promising solution for addressing environmental concerns associated with the overuse of paclobutrazol, a widely applied plant growth regulator in the cultivation of crops such as O. japonicus, potatoes, and mangoes (Kumar et al., 2021a; Deng et al., 2024). The excessive application of paclobutrazol has raised significant environmental issues, including detrimental effects on soil enzyme activities, nutrient cycling, and the accumulation of pesticide residues in agricultural products (Mir et al., 2022; Solangi et al., 2024). In light of these challenges, various strategies have been investigated to reduce paclobutrazol residue and enhance environmental sustainability. For instance, Klebsiella pneumoniae strain M6 has been identified as a potent biocontrol agent and paclobutrazol-degrading microorganism (Deng et al., 2024), while Pseudomonas putida strain T7 has shown promise as an effective bio-agent for remediating paclobutrazol-contaminated soils (Kumar et al., 2021b).

Environmental factors also play a critical role in paclobutrazol degradation, with studies indicating that degradation rates can be significantly enhanced in open-field soils compared to controlled conditions (Wu et al., 2013). While several methods, such as cold plasma and ultrasonication, have been explored for degrading pesticide residues in food products (Mir et al., 2022), few studies have adequately focused on using microbial agents to improve soil health and increase O. japonicus production while simultaneously degrading paclobutrazol residue.

In 2022, we isolated and purified various microorganisms from soil samples collected in the O. japonicus cultivation region of Santai County, Mianyang City, Sichuan Province, China. Through spread plate and streak plate methods, and subsequent identification via PCR and phylogenetic analysis, we identified MYSH as a key strain capable of degrading paclobutrazol residue, as confirmed by carbohydrate utilization tests and gas chromatography (Ding et al., 2023). Our findings indicated that, under natural conditions, the degradation rate of paclobutrazol in soil was only 50.72% over one year, consistent with prior research (Nørremark & Andersen, 1990). However, with the application of MYSH, the degradation rate significantly improved (p < 0.05), reaching 75.18% within five months (Fig. 2).

To sum up, MYSH presents considerable potential for agricultural applications, offering an innovative and sustainable solution for remediating contaminated soils. The large-scale use of MYSH could yield economic benefits through reduced fertilizer and pesticide expenditures, as well as the creation of revenue streams from MYSH-based products. Moreover, MYSH can be seamlessly integrated into existing agricultural practices, complementing other sustainable methods to enhance overall environmental sustainability.

Conclusions

This study evaluated the effects of the novel microbial agent MYSH on soil paclobutrazol residue, soil enzyme activities, and the production of O. japonicus. The results demonstrated that MYSH significantly reduced paclobutrazol residue in soil, enhanced soil enzyme activities, and improved both the yield and dry matter ratio of O. japonicus. Based on the analysis of HP, MYSH was identified as the primary factor influencing yield and dry matter ratio. The PLS-PM revealed that MYSH’s effects were mediated through the degradation of paclobutrazol residue, which positively impacted soil enzyme activities and ultimately crop productivity. These findings suggest that MYSH is a promising solution for sustainable agriculture, offering both environmental remediation and crop enhancement. As a microbial-based approach to mitigate agrochemical contamination and improve crop yield, MYSH can contribute to more eco-friendly farming systems. For future research, specific next steps include conducting field trials across different soil types and climatic conditions to assess MYSH’s robustness and adaptability, investigating its long-term effects on soil health, expanding research to evaluate MYSH’s efficacy on a wider range of crops and agrochemical residues, exploring the molecular mechanisms of paclobutrazol degradation by MYSH, and studying potential synergistic effects of MYSH with other sustainable agricultural practices. These efforts will help optimize MYSH application strategies and fully realize its potential in sustainable agriculture.

Supplemental Information

Supplemental Information 1 Raw data for soil paclobutrazol residue, soil enzyme activities, and dry weight and fresh weight of Ophiopogon japonicus tuberous roots

Note: MYSH represents Micrococcus yunnanensis strain HY001.

Supplemental Information 2 Hierarchical partitioning using the “glmm.hp” package and partial least squares path modeling using the “plspm” package in R version 4.4.0

MYSH represents Micrococcus yunnanensis strain HY001. PR represents soil paclobutrazol residue. SA represents soil sucrase activity. APA represents soil alkaline phosphatase activity. DR represents drying rate.

The author thanks all members of the Mianyang Institute for Food and Drug Control for their help.

Additional Information and Declarations

Competing Interests

Author Contributions

Patent Disclosures

Data Availability

Jianguo Ding is employed by Mianyang Xintengyuan Traditional Chinese Medicine Co., Ltd. We made use of patented material in the study (Micrococcus yunnanensis strain HY001 and its application in the degradation of paclobutrazol: China Patent 202310693924.3[P]. 2023).

Xiaojun Jiang performed the experiments, analyzed the data, prepared figures and/or tables, authored or reviewed drafts of the article, and approved the final draft.

Huadong Wang conceived and designed the experiments, authored or reviewed drafts of the article, and approved the final draft.

Yi Huang conceived and designed the experiments, prepared figures and/or tables, and approved the final draft.

Hong Jin conceived and designed the experiments, authored or reviewed drafts of the article, and approved the final draft.

Jianguo Ding performed the experiments, prepared figures and/or tables, and approved the final draft.

Lin Ma performed the experiments, prepared figures and/or tables, and approved the final draft.

Lei Zheng analyzed the data, prepared figures and/or tables, and approved the final draft.

The following patent dependencies were disclosed by the authors:

Micrococcus yunnanensis strain HY001 and its application in the degradation of paclobutrazol: China Patent 202310693924.3[P]. 2023.

The following information was supplied regarding data availability:

The code and raw data are available in the Supplemental Files.

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
