# Peer review of "A novel microbial agent reduces soil paclobutrazol residue, enhances enzyme activities and increases Ophiopogon japonicus production"

_PeerJ, doi:10.7717/peerj.19008_

## Round 0.1 · original submission · Minor Revisions

Make minor revisions, taking into account reviewers' comments. Pay attention to attached files.

Reviewer 1 ·

Basic reporting

The manuscript is generally well-written, with clear and professional English throughout. However, some sections could be enhanced for clarity, precision, and fluency. The introduction effectively contextualizes the research and highlights the significance of addressing paclobutrazol contamination. The literature review is comprehensive but could benefit from more recent citations to strengthen its relevance.
• Simplify and streamline complex sentences in the introduction (lines 56-75).
• Correct minor grammatical errors (lines 113-121).

Experimental design

The study addresses a knowledge gap by investigating Micrococcus yunnanensis strain HY001 (MYSH) as a bioremediation agent. The research question is well-defined, and the methodology is described sufficiently to allow replication. Nonetheless, certain methodological elements could be clarified.
• Provide a reference to the specific environmental details of the experimental site, such as the climate and the soil conditions (lines 131-133).
• Indicate the cultural operations carried out during cultivation (e.g. pesticide spraying, fertilization) (lines 133-139).

Validity of the findings

The findings are supported by rigorous statistical analysis and a comprehensive presentation of results. However, certain statistical methods, such as partial least squares path modeling (PLS-PM), would benefit from additional explanation for readers less familiar with advanced statistical techniques.
• Explain the PLS-PM methodology in more detail (lines 189-191).

Additional comments

The manuscript demonstrates the potential of MYSH for advancing agricultural sustainability through environmental remediation. The manuscript is well organized and logically structured. Expanding the discussion on broader agricultural applications would further enhance its impact.
• Discuss the potential scalability and commercial viability of MYSH.
• Compare MYSH’s performance with other microbial agents referenced in the literature.
• Expand the conclusion to outline specific next steps for future research and field applications.

Reviewer 2 ·

Basic reporting

Minor revisions have been requested for the manuscript. It is appropriate to publish after revision. Suggestions and corrections are indicated on the text.

Experimental design

good

Validity of the findings

good

Additional comments

no comment

Annotated reviews are not available for download in order to protect the identity of reviewers who chose to remain anonymous.

---

## Round 0.2 · accepted · Accept

Your manuscript was accepted after your revisions.

Reviewer 1 ·

Basic reporting

The revised manuscript demonstrates significant improvement in clarity and structure. The authors have addressed most of the issues highlighted in the initial review:

-The introduction has been refined, with technical terms such as "Hierarchical Partitioning (HP)" and "Partial Least Squares Path Modeling (PLS-PM)" now clearly defined.

-The grammatical and stylistic errors noted previously (e.g., lines 112-122) have been corrected.

-New and relevant literature citations have been added, strengthening the contextual foundation of the study.

Experimental design

The methodology is now described in sufficient detail, enabling reproducibility.

Key improvements include:

-Inclusion of specific environmental details of the experimental field (e.g., climate, soil type).

-A detailed explanation of the randomization process in the experimental design.

-Justification for the choice of soil enzyme activities, aligning them with study objectives.

Suggestions:

-While the revisions address randomization, a flowchart or diagram illustrating the experimental design would provide additional clarity.

-Consider explaining the rationale for the use of specific fertilizers in relation to the microbial agent’s effects.

Validity of the findings

The statistical analysis is now well-explained and thorough. The revised manuscript addresses previous concerns about the PLS-PM methodology and provides clear justifications for its use. The authors have also included additional visual aids, which enhance the interpretation of key findings.

Suggestions:

-Ensure all visual aids (e.g., figures and tables) have detailed captions for standalone comprehension.

-Discuss any potential biases in soil sampling that might impact the validity of the results.

Additional comments

The revisions substantially improve the manuscript’s overall quality. The discussion now better contextualizes the findings within broader agricultural applications, addressing scalability and sustainability of MYSH.

Suggestions:

-Expand the discussion further by comparing MYSH with other microbial agents cited in the literature.

-The conclusions effectively summarize the findings; however, emphasizing the practical implications for farmers and agricultural policymakers would increase the manuscript's impact.